# The Gestational Effects of Maternal Appetite Axis Molecules on Fetal Growth, Metabolism and Long-Term Metabolic Health: A Systematic Review

**DOI:** 10.3390/ijms23020695

**Published:** 2022-01-09

**Authors:** Angelos Dimas, Anastasia Politi, George Papaioannou, Thomas M. Barber, Martin O. Weickert, Dimitris K. Grammatopoulos, Sudhesh Kumar, Sophia Kalantaridou, Georgios Valsamakis

**Affiliations:** 13rd University Department of Obstetrics & Gynecology, Attikon University Hospital, Medical School of Athens, Ethnikon and Kapodistriakon University of Athens, 12462 Athens, Greece; papaioannougk@gmail.com (G.P.); skalanta@med.uoa.gr (S.K.); 2Nephrology Department, University Hospital of Ioannina, Stavros Niarchos Ave., 45500 Ioannina, Greece; anastasia.k.politi@gmail.com; 3Warwickshire Institute for the Study of Diabetes, Endocrinology and Metabolism, University Hospitals Coventry and Warwickshire, Clifford Bridge Road, Coventry CV2 2DX, UK; T.Barber@warwick.ac.uk (T.M.B.); martin.weickert@uhcw.nhs.uk (M.O.W.); Sudhesh.Kumar@warwick.ac.uk (S.K.); 4Institute of Precision Diagnostics and Translational Medicine, Pathology, University Hospitals Coventry and Warwickshire (UHCW) NHS Trust, Coventry CV2 2DX, UK; D.Grammatopoulos@warwick.ac.uk (D.K.G.); gedvalsamakis@yahoo.com (G.V.); 5Reproductive Endocrinology Unit, 3rd University Department of Obstetrics & Gynecology, Attikon University Hospital, Medical School of Athens, 12462 Athens, Greece; 62nd University Department of Obstetrics & Gynecology, Aretaieion University Hospital, Medical School of Athens, Ethnikon and Kapodistriakon University of Athens, 12462 Athens, Greece

**Keywords:** ghrelin, leptin, GLP-1, IGF-1, gut hormones, maternal hyperphagia, fetal adiposity

## Abstract

Increased maternal food intake is considered a normal pregnancy adjustment. However, the overavailability of nutrients may lead to dysregulated fetal development and increased adiposity, with long-lasting effects on offspring in later life. Several gut-hormone molecules regulate maternal appetite, with both their orexigenic and anorectic effects being in a state of sensitive equilibrium. The aim of this manuscript is to systematically review literature on the effects of maternal gut-hormone molecules on fetal growth and metabolism, birth weight and the later metabolic health of offspring. Maternal serum ghrelin, leptin, IGF-1 and GLP-1 appear to influence fetal growth; however, a lack of consistent and strong correlations of maternal appetite axis hormones with birth weight and the concomitant correlation with fetal and birth waist circumference may suggest that these molecules primarily mediate fetal energy deposition mechanisms, preparing the fetus for survival after birth. Dysregulated intrauterine environments seem to have detrimental, sex-dependent effects on fetal energy stores, affecting not only fetal growth, fat mass deposition and birth weight, but also future metabolic and endocrine wellbeing of offspring.

## 1. Introduction

According to the Barker et al. hypothesis of developmental origins of health and disease (DOHaD), fetal development is particularly vulnerable during the antenatal period [1,2]. This process of “fetal developmental programming” can have a long-lasting or even permanent impact on the offspring’s health trajectory through structural and functional disorders within the developing fetus [2]. Studies of pregnancies during the Dutch famine suggest that in-utero fetal exposure to malnutrition was associated with a variety of adverse metabolic phenotypes in offspring, such as elevations in BMI and serum cholesterol and worsening insulin resistance. Exposure to malnutrition during mid or late gestation resulted in lower birth weight and decreased head circumference [3,4]. Additionally, recent studies suggest that intrauterine growth restricted (IUGR) fetuses, regardless of cause, have a higher risk of adverse fetal bone growth, shorter stature in childhood and increased susceptibility to the development of osteoporosis due to the effects of adverse fetal programming on bone growth and of mineralization and adipocytokines on bone metabolism [2].

Although increased food intake is considered a normal adjustment during pregnancy, more than 40% of pregnant women exceed the current guidelines for optimal weight gain during pregnancy [5]. The overavailability of nutrients to the fetus can have adverse effects on fetal birth weight, increase adiposity through central allocation of fat deposition and facilitate fetal metabolic programming, ultimately increasing the lifetime risk of obesity and metabolic syndrome [6]. Excessive nutrient availability can also develop in obese pregnant women with excessive gestational weight gain and in some mothers with diabetes mellitus [6]. Various maternal molecules seem to influence fetal growth and metabolism. Recently, maternal molecules implicated in bone metabolism such as sclerostin correlated positively with fetal abdominal circumference and birth weight. Further, maternal sRANKL (the soluble form of receptor activator of nuclear factor kappa-Β ligand) was shown to correlate positively with fetal abdominal subcutaneous fat thickness, sagittal abdominal diameter and abdominal circumference [7]. Maternal adipocytokines and hormones such as visfatin, adiponectin and C peptide also associate (both directly and indirectly) with fetal growth and adiposity in various studies [8,9,10,11]. Lastly, reports suggest that material serum insulin and IGF-1 levels play an important role in fetal intrauterine growth and development [12,13]. Maternal IGF-1 is positively associated with neonatal ponderal index (PI)—as calculated by the formula weight(g) × 100/length(cm^3^)—mid-upper arm circumference and abdominal circumference in newborns and fetuses during the second and third trimesters [14,15]. Furthermore, maternal IGF-1 binding protein inversely correlated with mid-upper arm circumference at birth [15].

Several gut-hormone molecules regulate maternal appetite during gestation through both orexigenic and anorectic effects. Ghrelin increases the desire to seek and consume food, promotes fat deposition and gluconeogenesis and diminishes energy expenditure through reductions in physical activity, thereby directly opposing the appetitive and metabolic effects of leptin [16,17]. Neuropeptide Y (NPY), the most potent brain orexigenic peptide, increases body weight and adiposity [18]. Conversely, peptide YY inhibits food intake and weight gain and opposes the effects of ghrelin on food intake and energy balance [19,20]. The incretin hormones, glucagon-like peptide-1 (GLP-1) and insulin-like growth factors 1 and 2 (IGF-1 and -2), generally have anorectic properties [15,21,22,23]. Finally, leptin (the main anorectic hormone) released by adipocytes when fat stores are in excess [16,24], inhibits food intake and regulates nutrient uptake [25].

There is a complex interplay between these forementioned molecules and the regulation of appetite and food intake behavior in humans. Furthermore, although maternal appetite and dietary patterns seem to play a pivotal role in fetal development, growth and metabolism during gestation [6,26], there is a paucity of literature regarding the role of appetite-regulating gut-hormone molecules during pregnancy and their effect on fetal growth. The aim of this study is to systematically analyze literature and summarize the effects of maternal gut-hormone molecules on appetite regulation, fetal growth and metabolism, birth weight and the metabolic health of offspring in later life.

## 2. Materials and Methods

Our aim was to explore the effects of maternal gut-hormone molecules during pregnancy on intrauterine fetal growth (including ultrasound-derived measures such as estimated fetal weight, femur length, tibia length, abdominal circumference and subcutaneous fat deposition), anthropometric measurements at birth (including birth weight, abdominal and waist circumference at birth) and the longer-term metabolic health of offspring. We used the medical search engines Scopus and PubMed. Our search was performed using the following keywords, separately or in combination: maternal appetite molecules, maternal appetite hormones, leptin, ghrelin, NPY, PPY, PP, GLP-1, IGF-1, cholecystokinin, oxyntomodulin, GIP, fetal growth, fetal metabolism, birth anthropometry, birth weight and offspring metabolism. We sought human studies, prospective cohort studies, systematic reviews and meta-analyses in English published in medical journals prior to 1 October 2021.

Amongst the 2840 published papers considered, 2587 were excluded during the screening phase because they were out of the review scope as they concerned only fetal-and not maternal-appetite axis hormones, animal models, or referred to gestational pathology, such as gestational diabetes mellitus or preeclampsia. The remaining 206 full-text papers after duplicate removal, were assessed for eligibility. Finally, 33 studies were selected that focused on maternal gut-hormone molecules influencing fetal growth, fetal metabolism, birth anthropometry, neonatal metabolism and the metabolic health of offspring in healthy pregnancies. The flow chart of the study is reported in Figure 1.

## 3. Results and Discussion

### 3.1. Maternal Appetite Axis Molecules in Pregnancy and Fetal Growth

There are only a few reports in the literature that address possible interactions between maternal appetite hormone levels and intrauterine fetal measurements, summarized in Table 1. In their study Kubota et al. found a significant positive relationship between maternal serum IGF-1 levels and fetal biparietal diameter during the second and third trimesters of pregnancy. However, no similar correlation was reported regarding serum IGF-2 and ultrasound indices [12]. IGF-2 shares structural similarity with IGF-1, exerting antiapoptotic, growth-regulating and mitogenic effects [28]. IGFs circulate bound to six types of high-affinity proteins (IGFBP 1–6) that regulate their bioavailability and bioactivity through proteolysis and phosphorylation [15,23]. Baldwin et al. found an inverse relationship between maternal IGFBP-1 during the second trimester (20–24 weeks) and several fetal dimensions assessed by ultrasound, such as biparietal diameter, abdominal circumference, femur length, tibia length and subcutaneous fat. This relationship was not observed, however, during the third trimester [29]. In a prospective cohort study of 574 healthy pregnancies, Walsh et al. [30] reported that ultrasound-estimated fetal weight at 32 weeks of gestation was related to maternal serum leptin levels at booking [30].

Lastly, in their prospective study, Ruiz-Palacios et al. found that maternal serum insulin levels at the beginning of the third trimester were associated with fetal abdominal circumference Z-score, further supporting the important role of maternal insulin in fetal development during the early stages of gestation [13,31]. In another prospective study, Valsamakis et al. reported a negative correlation of maternal fasting plasma active GLP-1 (known for its anorectic actions) with fetal abdominal circumference measurements during the second trimester, whereas no other correlation was found between ultrasound measures of fetal development and serum levels of total GIP, active ghrelin and total PYY [22]. In the same study, maternal fasting plasma active GLP-1 level during the first trimester was also found to be the best negative predictor of fetal abdominal circumference during the second trimester. Thus, according to these preliminary studies, it seems that maternal appetite axis molecules are mostly implicated in the deposition of fetal adipose tissue and energy storage processes, rather than fetal growth.

### 3.2. Maternal Appetite Molecules and Anthropometrics at Birth

Over the years, studies have revealed a clear effect of maternal appetite axis hormones on fetal anthropometrics and placental weight at birth. Numerous studies have endeavored to correlate neonatal birth weight with maternal hormonal biomarkers to facilitate early detection of large for gestational age fetuses (LGAs). Amongst them, leptin and ghrelin are the most extensively studied maternal appetite-regulating molecules. The majority of studies were conducted during the late second or early third trimesters, which is partly** justified as hormonal levels can be measured during the almost universal screening of pregnancies for gestational diabetes mellitus (GDM) at that gestational age (Table 2).

Valsamakis et al., in their prospective study including 80 uncomplicated singleton pregnancies, demonstrated that serum levels of maternal activated ghrelin during the second and third trimesters showed a positive correlation with the newborn’s waist circumference at birth and during the third trimester—a negative correlation with percent total neonatal body fat. In fact, maternal serum ghrelin levels during the second trimester were the best positive predictor of waist circumference at birth, compared to other molecules studied [32]. Although no correlation was found between maternal serum ghrelin levels and birth weight, waist circumference at birth is considered and indicator of liver volume and visceral fat deposition, reflecting the newborn’s energy stores. Moreover, maternal serum leptin levels during the third trimester correlated negatively with neonatal waist circumference [32] suggesting a possible equilibrium between the antagonistic effects of maternal leptin and ghrelin levels. Furthermore, a prospective study by Chiesa et al. including a total of 153 singleton pregnancies, concluded that maternal serum ghrelin levels were positively correlated with neonatal head circumference at birth after adjusting the data for sex and gestational age. Nevertheless, mothers of LGA infants did not differ significantly in their serum ghrelin levels compared to mothers of full-term AGA infants [33].

In humans, ghrelin levels in umbilical cord blood have been found to negatively correlate with birth weight in both singleton and twin term pregnancies [34,35]. Similarly, elevated levels of ghrelin are found in the umbilical cord blood of fetuses with growth retardation whose mothers use tobacco, possibly as part of a compensatory mechanism in an environment of reduced nutrient uptake by the placenta [17]. In contrast, the role of maternal plasma ghrelin levels in developing fetuses and in birth weight has not yet been fully elucidated. Studying 36 uncomplicated pregnancies in all three trimesters prospectively, Saylan et al. concluded there was no relationship between maternal ghrelin levels and fetal birth weight or placental weight, although all neonates in the study had a birth weight within normal limits [36]. Similarly, no effect of maternal ghrelin on birth weight or placental weight was demonstrated in a study of 85 pregnancies by Bouhours-Nouet et al., even though maternal serum levels of ghrelin were found to be significantly increased during fasting, possibly signaling a reduction in available nutrients [17]. From the above studies, it seems possible that maternal ghrelin levels may be of greater significance to the enhancement of fetal energy reserves through fetal visceral adipose tissue accumulation, rather than fetal anthropometrics. Regarding maternal GLP-1, and despite a scarcity of data, Valsamakis et al. also reported a negative correlation between fasting plasma active GLP-1 in the second trimester and birth weight, with researchers suggesting its possible role as part of a compensatory physiological mechanism to pregnancy-related maternal hyperglycemia and insulin resistance [22].

In a prospective study of 177 pregnancies, Perichart-Perera et al. demonstrated a positive relationship between early pregnancy maternal serum leptin levels and birth weight [37]. However, this was only true for gravidas with a prepregnancy BMI within normal limits, after weighting results for gestational weight gain. The authors suggested the existence of different biological mechanisms between normal BMI pregnancies and those complicated with maternal overweightness or obesity [37]. Conversely, Retnakaran et al. showed in a prospective observational cohort study that included 472 pregnancies with leptin measurement taken during oral glucose tolerance testing (OGTT), which after adjusting for co-factors—including BMI and gestational weight gain—showed that maternal leptin levels were negatively correlated with birth weight [38]. Furthermore, in a prospective study by Kos et al., only 12 pregnancies with normal OGTT were studied, with comparable BMI and gestational weight gain. The authors also reported a negative correlation between maternal free serum leptin concentrations at 30 weeks of gestation and birth weight, a correlation also observed in an equal number of gravidas suffering from type 1 diabetes mellitus in the same study [39]. Results from yet another prospective cohort study reported by Misra et al. were similar. They studied 286 pregnancies and measured maternal serum leptin levels during all three trimesters, and failed to identify a significant relationship with birth weight [40]. Discrepancies observed in the above studies may stem from differences in study design (for example, while both studies aimed to compare leptin levels between normal BMI and overweight or obese mothers, Perichard et al. excluded macrosomic embryos from analysis, and Misra et al. included pregnancies complicated with hypertensive disorders), small sample sizes, or time points of leptin level assessment in maternal serum during pregnancy (first versus late second or early third trimesters).

The data outlined here regarding maternal serum leptin levels, unadjusted for maternal BMI, gestational weight gain or both, should be carefully evaluated, as both maternal BMI and gestational weight gain significantly affect patterns of leptin secretion during pregnancy, possibly leading to different biological effects and constituting strong independent variables affecting neonatal birth weight [40,41]. To illustrate this point, in their prospective cohort study including 537 pregnancies, Walsh et al. [30] showed that maternal serum leptin levels during early pregnancy correlate with both EFW at 34 weeks of gestation and neonatal birth weight. This correlation was not verified using maternal serum leptin levels at 28 weeks of gestation. However, these forementioned results did not take into account maternal gestational weight gain [30]. Furthermore, Shroff et al. [42], studying a large subcohort sample of 1304 pregnancies from the “POUCH” study between the 16th and 27th weeks of gestation, also found that the log of elevated serum leptin levels predicted an LGA neonate, but these data also lacked any adjustment for gestational weight gain. However, after adjusting for maternal pre-pregnancy BMI, the above correlation attenuated, nevertheless reaching statistical significance only in preterm births (<37 weeks) [42]. A positive relationship between the logarithm of maternal serum leptin concentration during the first trimester and LGA birth as well as birth weight was also found in a prospective cohort study of Farias et al., which included 199 gravidas [43]. Nevertheless, regarding neonatal birth weight, statistical significance was not reached when data were adjusted for maternal BMI. Similarly, a prospective study of 278 pregnancies during OGTT (24 to 29 weeks) by Verhaeghe et al. found that women with elevated serum leptin levels had a tendency of giving birth to neonates with higher-than-normal weights; however, results did not reach statistical significance. The authors concluded that maternal serum leptin level measurements in uncomplicated pregnancies during OGTT are not clinically useful in predicting neonatal birth weight [44].

No relationship was reported between maternal serum leptin and birth weight by Clausen et al. [41]. In a subsequent cohort study in 2005, Verhaeghe et al. [45] included pregnancies complicated with hypertensive disorders or preeclampsia, as well as those with abnormal OGTT results. In their study sample, maternal serum leptin levels significantly and negatively correlated with neonatal birth weight. In preeclamptic pregnancies (PE), leptin levels were found to change significantly, especially in clinical evident cases. Furthermore, the leptin expression profile seems to vary depending on time of PE onset, with higher levels reported in early- versus late-onset PE [45], thus complicating analysis. As a result, inclusion of complicated pregnancies is likely to have significantly affected the study results. Nevertheless, from a clinical perspective, maternal serum leptin levels during OGTT explained only 1.8% of birth weight variation [45]. Studies mentioned above are summarized in Table 3.

Various other studies of maternal leptin levels in relation to neonatal birth weight are available in the literature, regarding macrosomic neonates in particular, that disregard the influence of significant variables such as maternal BMI and gestational weight gain on birth weight.

In some of these, maternal plasma leptin levels during the third trimester and during labor did not correlate with birth weight, nor did they differ significantly between mothers of LGA and those of AGA or SGA neonates [24,46,47,48,49]. Lastly, in a prospective study of 49 pregnancies, birth weight z-score negatively correlated with maternal serum levels of IGFBP-1 during the third trimester. The authors postulated that lower maternal serum IGFBP-1 levels contribute towards the development of macrosomic neonates through increased IGF-1 bioavailability [50]. Controversies in the literature and non-definitive study results suggest that, ultimately, maternal gut hormones may not play an important role in the determination of neonatal birth weight.

### 3.3. Maternal Appetite Molecules and Fetal Metabolism

Current evidence suggests that maternal conditions such as obesity, nutritional status and high dietary intake may contribute towards fetal metabolic alterations, through facilitating placental nutrient transport, mediated by circulating maternal serum insulin and leptin in overweight and obese mothers [50,51]. However, the literature contains only a few studies of amniotic fluid or cord blood metabolic indices in relation to maternal appetite-regulating gut hormones. In one of these, in 80 pregnancies a negative correlation between maternal serum activated ghrelin levels during the third trimester with cord blood insulin levels was reported [32]. Other study results of 574 uncomplicated pregnancies suggest that maternal leptin in early pregnancy and at 28 weeks correlate with fetal insulin resistance, as assessed by C peptide levels in cord blood (Table 4). The authors concluded that maternal leptin levels may be clinically useful as a biomarker of intrauterine insulin resistance, independent of maternal BMI [30]. A summary of important findings of studies so far is depicted in Figure 2.

The composition of amniotic fluid in early pregnancy is similar to maternal interstitial fluid, due to resorption through fetal membranes. This process is most likely regulated by the placenta. However, from 20 weeks of gestation on, fetal urine and lung secretions have a predominant effect on its composition [52]. Therefore, amniotic fluid reflects maternal and fetal health status simultaneously, and utilizing metabolomics, it can be used to assess various parameters of fetal development and metabolic indices [52,53]. In addition, there are reports that cord blood metabolites are associated with neonatal anthropometrics and body composition at birth [54]. Furthermore, studies suggest that pregnancies with maternal endocrine disorders- such as gestational diabetes mellitus or hyperglycemia, result in altered cord blood metabolism [55], whereas maternal metabolites of both healthy mothers and mothers with hypercholesterolemia were associated with several cord blood metabolites in a sex-dependent manner [56]. As a result, cord blood seems to be a mixture of maternal and fetal metabolism, along with nutrients transferred through the placenta.

More research is required to gain better insight into fetal intrauterine metabolic alterations influenced by maternal overnutrition and obesity. Nevertheless, it seems that circulating maternal molecules can influence the metabolism of the feto-placental unit despite the presence of the placental barrier [57]. It is well known that the placenta forms the interface between the fetus and its mother and is the organ that oxygen and nutrients have to pass through in order to be delivered in fetal circulation. However, the placenta also acts as a selective barrier in order to minimize fetal exposure to molecules, such as maternal hormones or even pathogens.

### 3.4. Maternal Appetite Axis Molecules during Pregnancy and Possible Metabolic or Endocrine Consequences in Offspring

In our literature review, we did not find any direct studies of maternal appetite axis molecules during gestation in relation to future metabolic or endocrine diseases in the offspring. However, it is well accepted that maternal over- or undernutrition, obesity and gestational weight gain in pregnancy have detrimental effects on fetal development and the later metabolic health of offspring. It is thus possible that maternal appetite axis hormones have a major impact on maternal eating behavior during pregnancy, influencing the offspring’s metabolic and endocrine health in later life. Increased maternal nutrient supply may result in an altered fetal gene expression profile through epigenetic modifications in utero. Indeed, studies have shown that maternal overnutrition increases fetal expression of adipogenic transcription factor PPARγ, lipoprotein lipase, adiponectin and adipose tissue-derived leptin, increasing the risk of obesity development for offspring in later life [58,59,60]. Recent advances in epigenetics suggest that obesity in pregnancy associates with the future development of metabolic syndrome in the offspring. Furthermore, neonatal and childhood obesity seems to originate from intrauterine nutritional status combined with other environmental exposures, such as maternal or fetal stress [4].

A wide range of lipid metabolism disorders have been observed in diabetic pregnancies, with serum lipid levels in mothers diagnosed with gestational diabetes influencing fetal development, as maternal dyslipidemia is likely to increase the bioavailability of lipids to the fetus [58,61,62]. The composition of the mother’s diet as well as her metabolic status may also influence lipid transportation across the placenta, both directly due to their increased concentration and indirectly, altering the level of oxidative stress and inflammatory status within the placenta [61,62]. A high-fat maternal diet can also affect the fetal hypothalamic control center of appetite and energy equilibrium. Studies have shown that a high fat intake early in pregnancy combined with maternal obesity can lead to changes in the expression of genes encoding the leptin receptor, POMC and neuropeptide Y in the newborn through epigenetic DNA alterations. These modifications likely contribute towards increased adiposity, overeating and insulin resistance within the offspring [58,63,64].

To summarize, maternal obesity during pregnancy resulting from increased appetite is characterized by the elevation of maternal appetite axis molecules such as leptin and insulin, and a pre-inflammatory state that may enhance maternal lipogenesis. These changes may ultimately result in an increased white adipose tissue mass and adipocyte hypertrophy within the offspring. The maternal nutritional environment and hyperphagia can alter the expression of fetal genes through epigenetic mechanisms, leading to obese phenotypes within the offspring [58].

### 3.5. Maternal–Fetal Sex–Dependent Biological Competition for Nutrients

Studies indicate that human placenta adapts its nutrient transport characteristics, attempting to balance fetal demand signals with maternal resource allocation during healthy pregnancies. An example of this is placental lactogen secretion, exerting a maternal anti-insulin effect, which raises blood glucose and free fatty acid concentrations, facilitating their transfer to the fetus [65]. However, an adequate supply of nutrients is needed to sustain the equilibrium between fetal and maternal needs. In the presence of maternal undernutrition and inadequate energy supply, a state of biological competition develops between the conceptus and its mother that may compromise the well-being of both [63,66]. This seems to be particularly true in young adolescent pregnancies and in gestations with short interpregnancy intervals that exhibit a suboptimal ability to support fetal growth [63,66]. Scholl et al. [64] found that young, still growing teens show a maternal leptin surge during the third trimester, which may reduce the maternal fat breakdown rate, resulting in less energy available for fetal growth. Ultimately, the allocation of metabolic fuels in young pregnant adolescents favors maternal over fetal growth, resulting in lower neonatal birth weight and higher maternal fat gain [66]. The Dutch famine showed that the two sexes respond differently to nutritional stress in utero. The placental area was found to reduce significantly more for male fetuses when compared to females in a retrospective cohort study of 860 pregnancies [67]. The researchers concluded that their results are consistent with those of Eriksson et al., who, having studied 2003 pregnancies retrospectively, suggested that male fetuses grow more rapidly in utero, investing energy and nutrients more in the brain but less in placental growth compared to females, jeopardizing their well-being in the case of compromised nutrition [68]. It also seems that the feto-placental growth of male fetuses is influenced more by maternal diet during pregnancy, while maternal metabolism and lifetime nutrition has a greater impact on the growth of female fetuses [68,69] (Table 5). However, in the literature, there is a lack of studies focusing on gender-specific differences of maternal gut hormones on fetal and neonatal growth and metabolism.

Dysregulated intrauterine environments, such as obesity and gestational diabetes, seem to have detrimental effects on fetal energy stores, affecting not only fetal growth, fat mass deposition and birth weight, but also future metabolic and endocrine wellbeing of offspring. Studies in both animal and human models have shown that disrupted maternal leptin and ghrelin secretion homeostasis may lead to the release of abnormal hypothalamic signals, thereby promoting a feeling of hunger, resulting in excessive food consumption and lipogenesis [70,71]. Researchers also reported that hyperinsulinemia decreases GLP-1 secretion from human intestinal L cells. Therefore, induced leptin resistance due to constant hyperleptinemia may account for decreased GLP-1 levels and increased appetite found in obese individuals [72]. Furthermore, recent findings suggest that excessive gestational weight gain is associated with increased maternal insulin and leptin levels [73,74]. Ultimately, maternal hyperinsulinemia, hyperleptinemia and proinflammatory states are linked with excessive nutrient placental transport to the fetus [62,71], which may enhance fetal lipogenesis, leading eventually to increased white adipose tissue mass, adipocyte hypertrophy or higher fat mass percentiles in the offspring [75].

The maternal nutritional environment and hyperphagia can also alter the expression of fetal genes through epigenetic mechanisms, such as the transcription factor PPARγ, leading to obese phenotypes [58]. As an example, high fat intake early in pregnancy combined with maternal obesity can lead to changes in the expression of genes in the fetus, including leptin receptor expression, POMC and neuropeptide Y mediated through epigenetic alterations. These changes are likely to lead to increased adiposity, overeating and insulin resistance in the offspring [58,63,64]. Further, placental gene expression patterns seem to vary significantly according to fetal gender. Preliminary data support the notion that gene alterations in male placentae may be part of a biological mechanism that enables the continued fetal growth in an adverse intrauterine environment [76].

## 4. Conclusions

Although the exact biological mechanisms involved remain elusive, maternal serum ghrelin, leptin, IGF-1 and GLP-1 appear to influence intrauterine fetal growth. Nevertheless, their effects need to be further clarified with additional prospective human studies. In the current literature, numerous studies correlate fetal development with several maternal serum hormonal biomarkers. However, significant heterogeneity in the methods used and the small sample sizes hinder direct comparisons between studies. It is nevertheless possible that enhanced maternal appetite during gestation may reflect not just a mother’s own nutrient supply needs as measured by ghrelin or other gut hormonal levels but also the rising energy needs of the developing fetus, ultimately affecting its future metabolic health. Maternal appetite-regulating hormones in normal pregnancies may mediate fetal energy deposition mechanisms, rather than enhance fetal or neonatal anthropometric measurements. Recent research suggests that fetal adipose tissue and energy stored in fat may serve not only as insulation for temperature variations between the in-utero and post-delivery periods, but also to prepare the developing fetus for survival outside the protective intrauterine environment. For example, it was proposed that it may also serve to sustain the large human brain by providing the energy needed through ketone bodies and contribute towards energy and molecular signals to mediate the normal development of neonatal immune functions [77,78].

We have confirmed a scarcity of studies addressing the possible effects of maternal appetite axis regulating molecules on fetal intrauterine growth as estimated by ultrasound indices, although reports suggest that some may play a role. We did not find studies regarding maternal appetite axis molecules in relation to fetal metabolic parameters in amniotic fluid or in cord blood. Given literature reports that correlate fetal or neonatal metabolites with various maternal molecules, it would be of no surprise if maternal appetite axis hormones influenced fetal metabolic indices in amniotic fluid or cord blood, but this is, to date, a knowledge gap. Exceptions to this are the negative correlation of maternal third trimester activated ghrelin levels with cord blood insulin levels reported by Valsamakis et al. and the reported correlation of cord blood C peptide levels with maternal leptin in early pregnancy and at 28 weeks reported by Walsh et al. However, other studies suggest that the cord blood lipidome and metabolome are sensitive to the intrauterine environment as confirmed in pregnancies complicated by obesity, gestational diabetes or hypercholesterolemia [79,80,81,82] as well as maternal dietary patterns [83]. Finally, we also highlight a lack of direct studies regarding maternal appetite axis molecules during gestation in relation to future metabolic or endocrine diseases in the offspring, even though their indirect effects through maternal obesity and excess gestational weight gain seem to be widely accepted.

Further research is required to elucidate the role of maternal appetite axis hormones during pregnancy and human fetal development, including their influence on fetal metabolism and metabolic/endocrine health in later life. Nevertheless, researchers should rely on large, prospective and carefully designed human studies (with significant statistical power) for those questions to be answered. Cutoffs used for categorizing neonates in birth weight groups should be homogeneous between studies to allow direct comparisons, while gestational ages at birth should also be considered using centiles for neonatal anthropometric parameters. Evidence in the literature of altered biological mechanisms of appetite regulation in cases of maternal obesity or overweightness suggest that maternal BMI, as well as maternal gestational weight gain, dietary patterns during pregnancy and even fetal gender, should be considered when designing future studies [40].

Moreover, maternal appetite-regulating molecules should be evaluated at various timepoints during gestation, as it seems that timing of fetal exposure plays a pivotal role in human fetal growth [22,30]. The widespread use of ultrasonography in obstetrics can also be further utilized to explore possible correlations between maternal gut hormone levels and fetal growth/fat deposition indices, as relevant studies are scarce in the literature. Lastly, future research should also focus on exploring possible correlations of maternal appetite axis hormonal levels with intrauterine fetal metabolism, given that evidence for such correlations exist. In order to study fetal metabolism, research could focus on cord blood samples or samples from amniocentesis taken during gestation when clinically indicated.

## Figures and Tables

**Figure 1 ijms-23-00695-f001:**
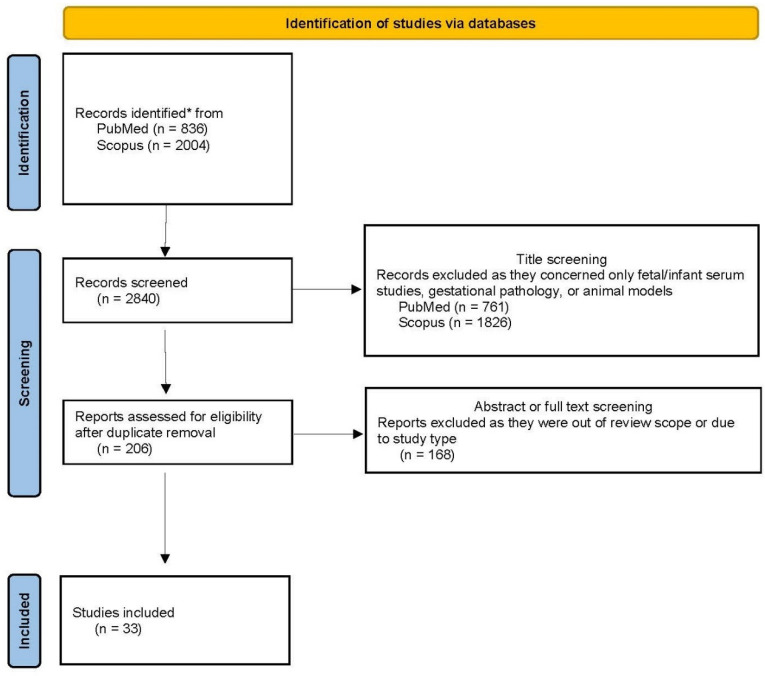
Flowchart of the study [27], * ((“maternal appetite molecules” OR “maternal appetite hormones” OR leptin OR ghrelin OR NPY OR PPY OR PP OR GLP-1 OR IGF-1 OR cholecystokinin OR oxyntomodulin OR GIP) AND (“fetal growth” OR “fetal metaboli *” OR “birth anthropometr *” OR “birth weight” OR “offspring metaboli *”)).

**Figure 2 ijms-23-00695-f002:**
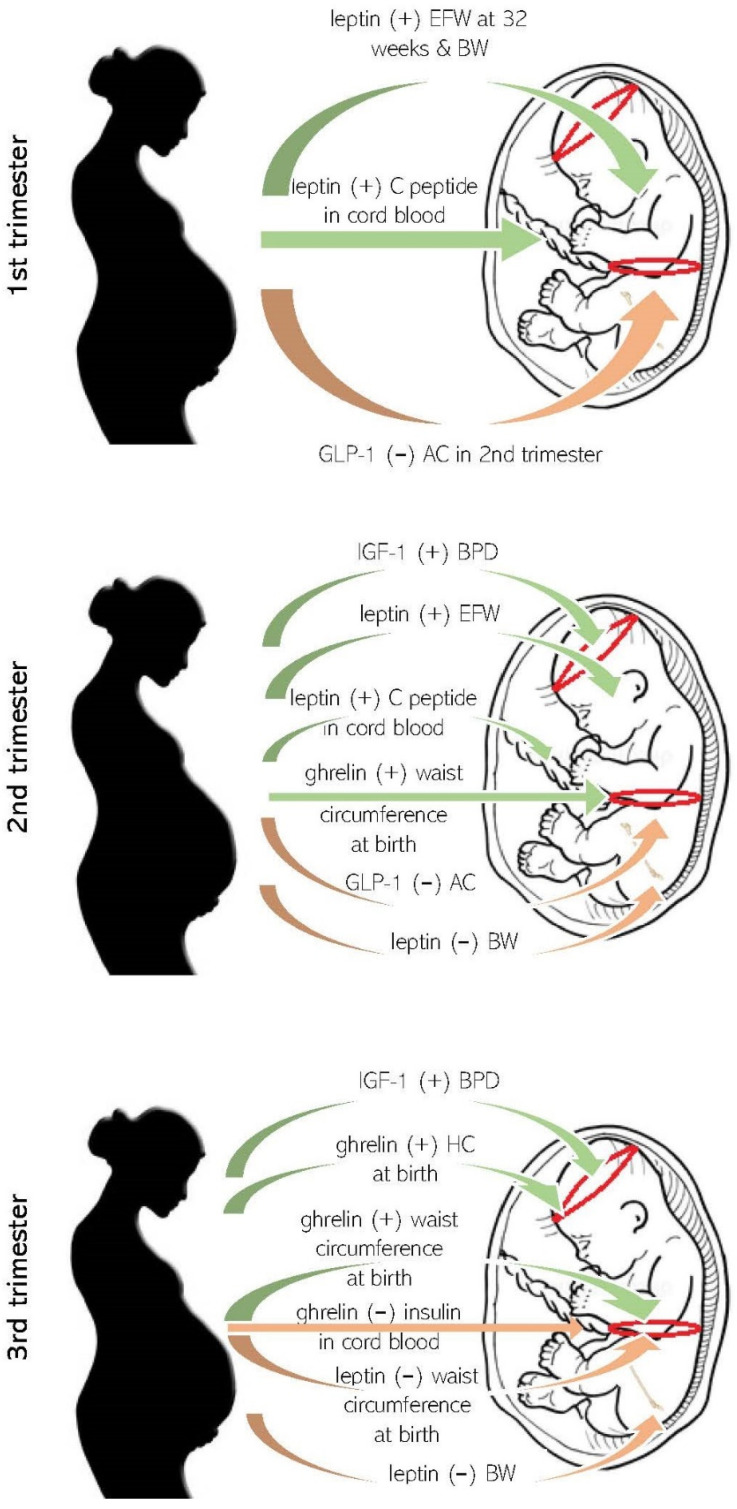
Schematic representation of important study results correlating maternal appetite-regulating molecules with fetal and neonatal growth and metabolic indices, in all three trimesters of human pregnancy. (+): positive correlation, (−): negative correlation, EFW: estimated fetal weight, AC: ultrasound estimated abdominal circumference, BPD: biparietal diameter, BW: birth weight.

**Table 1 ijms-23-00695-t001:** Studies of maternal molecules implicated in appetite regulation in relation to intra-uterine fetal growth indices. * BPD: biparietal diameter, ** AC: abdominal circumference, *** GDM: gestational diabetes mellitus.

Studies	Sample (N)	Maternal Molecule Studied	Trimester Studied	Main Outcome	Comments
Kubota et al.	52	IGF-1 IGF-2	Second and third	Positive correlation to fetal BPD * in second and third trimester (r = 0.606, *p* < 0.001) No correlations	
Baldwin et al.	200	IGFBP-1	Second (20–24 weeks)	Inverse correlation to fetal BPD_1_, AC_2_, femur_3_ and tibia_4_ length and subcutaneous fat_5_ (r_1_ = 0.319, *p* < 0.000, r_2_ = 0.257, *p* < 0.005, r_3_ = 0.288, *p* < 0.005, r_4_ = 0.243, *p* < 0.005, r_5_ = 0.326, *p* < 0.005)	No correlation found in third trimester/IGFBP-1 also inversely related with birth weight (r = 0.185, *p* < 0.05)
Walsh et al.	574	leptin	Early gestation and at 28 weeks	Correlation to estimated fetal weight at 32 weeks (r = 0.16, *p* < 0.001) and (r = 0.12, *p* = 0.008)	Early pregnancy leptin also correlated with neonatal birth weight (r = 0.14, *p* = 0.001)
Ruiz-Palacios et al.	68	insulin	Early third	Association to third trimester fetal AC ** (r = 0.266, *p* = 0.025)	Included GDM *** pregnancies
Valsamakis et al.	55	Active GLP-1 Total GIP Active ghrelin Total PYY	First	Negative correlation to fetal AC in second trimester (r = −0.55, *p* = 0.034) No correlations found	Active GLP-1 was the best negative predictor of second trimester AC

**Table 2 ijms-23-00695-t002:** Important study aspects of maternal molecules implicated in appetite regulation, in relation to neonatal anthropometric measurements at birth.

Studies	Sample (N)	Maternal Molecule Studied	Trimester Studied	Main Outcome	Comments
Valsamakis et al.	80	Activated ghrelin	Second and third	Positive correlation with neonatal waist circumference at birth (second: r = 0.75 and third: r = 0.70, *p* < 0.001 with *p* < 0.001) negative correlation with percent total body fat (r = −0.94, *p* < 0.001)	Ghrelin levels during second trimester were the best positive predictor of birth waist circumference, no relation to birth weight
		Leptin	Third	Negative correlation with neonatal waist circumference (r = −0.81, *p* < 0.001)	
		Active GLP-1	Second	Negative correlation with birth weight (r = −0.40, *p* = 0.03)	
Chiesa et al.	153	ghrelin	During labor	Positive correlation with head circumference at birth (B = 0.45 95% CI: = 0.17, 0.73, *p* < 0.01)	
Saylan et al.	36	ghrelin	All three trimesters	No relation to birth or placenta weight	All neonates had birth weight within normal range
Bouhours-Nouet et al.	85	ghrelin	During labor	No correlation to birth or placenta weight	
Valsamakis et al.	55	Activated GLP-1	Second	Negative correlation with birth weight (r = −0.50, *p* = 0.040)	
Perichart-Perera et al.	177	leptin	Early first	Positive correlation with birth weight (B = 0.007 95% CI: 0.002, 0.011, *p* = 0.005)	Valid only in normal maternal BMI pregnancies, excluded macrosomic neonates
Retnakaran et al.	472	leptin	Late second to early third	Negative correlation with birth weight Adj. OR (95% CI) = −3.92 (−6.23 to −1.60)	Leptin was found to be a significant negative predictor of birth weight and large-for-gestational-age neonate
Kos et al.	12	Free leptin	30 weeks of gestation	Negative correlation with birth weight (r = −0.63, *p* < 0.05)	Same negative correlation in type 1 diabetes mellitus pregnancies
Misra et al.	286	leptin	All three trimesters	No correlation with birth weight	Included pregnancies complicated with hypertensive disorders

**Table 3 ijms-23-00695-t003:** Studies of maternal leptin levels in relation to neonatal anthropometric measurements at birth, with their data not adjusted to maternal BMI, gestational weight gain or both. * EFW: estimated fetal weight, ** SGA: small for gestational age, *** AGA: appropriate for gestational age.

Studies	Sample (N)	Trimester Studied	Main Outcome	Comments
Walsh et al.	537	Early gestation	Correlation with EFW * at 34 weeks (β = 0.16, *p* = 0.02) and neonatal birth weight (r = 0.14, *p* = 0.001)	No similar significant correlations at 28 weeks of gestation
Shroff et al.	1304	Second	Elevated leptin predicts an LGA neonate	Included cases with gestational pathology/After adjusting data for maternal BMI, the correlation attenuated but remained significant in preterm births
Farias et al.	199	First	Positive correlation with birth LGA birth (intercept OR = 3.88; 95% CI: 1.49 to 10.09; *p* = 0.005)	Maternal log leptin in 1st trimester was also correlated with birth weight, but when model included data adjusted for maternal pre-pregnancy BMI, statistical significance was not reached
Verhaeghe et al.	278	Late second	Mothers with elevated leptin levels were more likely to give birth to an obese/overweight neonate	Authors stated that leptin levels measurement had no clinical use
Clausen T. et al.	2050	Early second	No correlation with birth weight	
Verhaeghe et al.	631	Late second	Negative correlation with birth weight (T = −4.10, *p* < 0.0001)	Included pregnancies with hypertensive disorders and/or preeclampsia and abnormal OGTT results
Lazo-de-la-Vega-Monroy et al.	60 20 with SGA **, 20 with AGA *** and 20 with LGA neonates	During labor	No relation with LGA neonates	
Horosz et al.	134 86 with GDM and 48 normal pregnancies	Early third trimester	No relation with AC or birth weight	
Ozdemir et al.	88	Late third trimester (>38 weeks)	No relation with birth weight	Maternal leptin levels were significantly higher in LGA group
Ökdemir et al.	84	During labor	No relation with anthropometrics at birth	
Papadopoulou et al.	85	Right after delivery	No correlation with placental or birth weight	No sex differences observed in leptin levels

**Table 4 ijms-23-00695-t004:** Studies of maternal appetite molecule levels in relation to fetal metabolic indices in cord blood at birth.

Studies	Sample (N)	Maternal Molecule Studied	Trimester Studied	Main Outcome	Comments
Valsamakis et al.	80	Activated ghrelin	Third	Negative correlation with insulin levels in cord blood (r = −0.82, *p* < 0.001)	Third trimester activated ghrelin was the best negative predictor of cord blood insulin levels
Walsh et al.	574	leptin	In early pregnancy and at 28 weeks	Correlated with C peptide levels in cord blood (β = 0.173, *p* = 0.004 and β = 0.115, *p* = 0.05 respectively)	Maternal leptin suggested to be utilized as a biomarker of fetal, intrauterine insulin resistance

**Table 5 ijms-23-00695-t005:** Studies supporting the presence of gender-specific effects of maternal nutritional status during gestation, on placental size and surface area.

Studies	Sample (N)	Indices Studied	Main Relevant Outcome	Comments
Van Abbelen et al.	860	Placental size at birth	Maternal undernutrition during gestation reduced placental surface area in men but not in women	Retrospective cohort study
Eriksson et al.	2003	Placental surface area at birth	Male fetuses grow more rapidly in-utero compared to females	Retrospective study of birth records/energy and nutrients are invested more in brain, but less in placental growth by male fetuses/maternal diet during pregnancy seems to influence growth of only male fetuses
Roseboom et al.	2414	Placental area and volume	Placentas of male fetuses had less surface area compared to their female counterparts	Retrospective study of birth records/Famine impaired normal placentation especially those in mid- late gestation

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
