# Peer review of "The Gestational Effects of Maternal Appetite Axis Molecules on Fetal Growth, Metabolism and Long-Term Metabolic Health: A Systematic Review"

_ijms, 2022, doi:10.3390/ijms23020695_

Round 1

Reviewer 1 Report

A very well written review on an interesting but challenging topic, with interventional studies being unlikely to be conducted in pregnant women, meaning that mainly correlations are being relied upon to infer causality. 

It is a little dense to read, but generally very well written. 

A summary figure, summarising the findings of the review would be beneficial.

Minor

Line 41 italicise in utero

Line 66 please explain neonatal ponderal index.

Line 194 although most readers will be familiar with IGF-1, can you add a sentence on IGF-2 and IGF BP for the general reader. 

Tables- can you add some description of the correlation observed for parameter where there is a significant relationship? 

Reviewer 2 Report

In This Manuscript the authors perform a systematic review of the literature on the effects of maternal gut hormones on fetal growth and metabolism, birth weight and offspring’s later metabolic health. This is an interesting topic and the authors clearly identify and review the key papers, but there are issues that must be considered in a revision.

-The key here is human studies, which were the ones selected for discussion. Yet, the authors start the Results and Discussion section with a review of mechanistic insights on this topic the relies also on animal data (although the species and models are not always clear). This first part does not belong here, but in a general introduction to the topic. The results should focus on the human data, and start on what is now section 3.2. Furthermore the different models and species discussed should always be clearly noted in the text.

-The sections are not uniform in style. Sections 3.4. and 3.6 (possibly not 3.5) are not backed up by the same type of very informative and reader-friendly tables of the previous sections, and also do not systematically list the numbers of patients involved in each study. This should be fixed and more Tables included so the paper is more homogeneous.

-The conclusions should also include the Authors more specific vision on what kind of studies would be most important to carry out in the future, focusing on what types of molecules/methodologies, with what numbers of patients. In short what are the most cogent problems and what they think would be a valid roadmap to reach more robust conclusions on at least some of the issues than the data currently allows beyond the mere, and very obvious notion that “more studies are needed”.

-Many references are not correctly mentioned throughout (first letter of the first name of authors is not needed).

-Is the term “gravidas" commonly used in the field?

Round 2

Reviewer 2 Report

The authors have addressed my previous concerns, and I have no further comments.